# A Method to Monitor the NAD^+^ Metabolome—From Mechanistic to Clinical Applications

**DOI:** 10.3390/ijms221910598

**Published:** 2021-09-30

**Authors:** Maria Pilar Giner, Stefan Christen, Simona Bartova, Mikhail V. Makarov, Marie E. Migaud, Carles Canto, Sofia Moco

**Affiliations:** 1Nestle Research, EPFL Innovation Park, H, 1015 Lausanne, Switzerland; mariapilar.giner@rd.nestle.com (M.P.G.); stefan.christen@rd.nestle.com (S.C.); simona.bartova@rd.nestle.com (S.B.); carles.cantoalvarez@rd.nestle.com (C.C.); 2Mitchell Cancer Institute, University of South Alabama, 1660 Springhill Avenue, Mobile, AL 36604, USA; mikhail.makarov@olonricerca.com (M.V.M.); mmigaud@southalabama.edu (M.E.M.); 3Olon Ricerca Bioscience, 7528 Auburn Road, Concord, OH 44077, USA; 4Division of Molecular and Computational Toxicology, Department of Chemistry and Pharmaceutical Sciences, Amsterdam Institute for Molecular and Life Sciences, Vrije Universiteit Amsterdam, De Boelelaan 1108, 1081 HZ Amsterdam, The Netherlands

**Keywords:** NAD^+^, metabolomics, mass spectrometry

## Abstract

Nicotinamide adenine dinucleotide (NAD^+^) and its reduced form (NADH) are coenzymes employed in hundreds of metabolic reactions. NAD^+^ also serves as a substrate for enzymes such as sirtuins, poly(ADP-ribose) polymerases (PARPs) and ADP-ribosyl cyclases. Given the pivotal role of NAD(H) in health and disease, studying NAD^+^ metabolism has become essential to monitor genetic- and/or drug-induced perturbations related to metabolic status and diseases (such as ageing, cancer or obesity), and its possible therapies. Here, we present a strategy based on liquid chromatography-tandem mass spectrometry (LC-MS/MS), for the analysis of the NAD^+^ metabolome in biological samples. In this method, hydrophilic interaction chromatography (HILIC) was used to separate a total of 18 metabolites belonging to pathways leading to NAD^+^ biosynthesis, including precursors, intermediates and catabolites. As redox cofactors are known for their instability, a sample preparation procedure was developed to handle a variety of biological matrices: cell models, rodent tissues and biofluids, as well as human biofluids (urine, plasma, serum, whole blood). For clinical applications, quantitative LC-MS/MS for a subset of metabolites was demonstrated for the analysis of the human whole blood of nine volunteers. Using this developed workflow, our methodology allows studying NAD^+^ biology from mechanistic to clinical applications.

## 1. Introduction

Many metabolic pathways depend on the redox couples, nicotinamide adenine dinucleotide NAD^+^ and reduced form NADH, and respective phosphorylated forms, nicotinamide adenine dinucleotide phosphate NADP^+^ and reduced form NADPH. NAD(H) and NADP(H) both serve as hydride carriers in the cell (donors and acceptors), but they act differently. In general terms, NAD(H) plays a crucial role in catabolism, energy metabolism (glycolysis, fatty acid oxidation, tricarboxylic acid cycle and electron transport chain), while NADP(H) is involved in anabolic reactions (biosynthesis of nucleic acids, fatty acids, cholesterol and steroid hormones) as well as in antioxidant response and detoxification processes [1]. Beyond their established biochemical roles in metabolism, these molecules have important cell signaling functions. NAD^+^ is a substrate for enzymes such as sirtuins, poly(ADP-ribose) polymerases (PARPs), sterile alpha and Toll/interleukin receptor motif containing protein (SARM) and ADP-ribosyl cyclases [2], which have significant roles in a variety of conditions, such as ageing, cancer and obesity [1,2,3,4]. Lower levels of NAD (either NAD^+^ or total NAD^+^ and NADH) have been associated with ageing [1,5], obesity [6], non-alcoholic fatty liver disease [7] and acute kidney injury [8]. Mechanistically, depletion of NAD levels is associated with mitochondrial dysfunction, and thereby influences the development of metabolic disorders [9]. A more detailed understanding is needed of the balance of consumption and biosynthesis of NAD(H) pools including precursors and catabolites.

In mammalians, NAD(P)(H) is synthesized by several pathways (Figure 1). The de novo biosynthesis synthesizes NAD^+^ from tryptophan in a multi-step pathway via the kynurenine pathway. The salvage pathway plays a more prominent role in synthesizing NAD^+^ in mammalians since it predominates in most cell types [2]. The precursors nicotinamide (Nam) and nicotinic acid (or niacin, NA) were proposed by Elvehjem [10] and Preiss and Handler [11], establishing two routes, an aminated and a non-aminated one, towards the synthesis of NAD^+^. In the 2000s, nicotinamide riboside (NR) [12] was proposed as an efficient NAD^+^ precursor with nicotinamide riboside kinase (NRK) as a rate-limiting step in producing nicotinamide mononucleotide (NMN) in the salvage pathway and eventually resulting in production of NAD^+^. The search for NAD^+^ precursors led to the discovery of the reduced form of nicotinamide riboside (or 1,4-dihydronicotinamide riboside, NRH) as a novel and extremely efficient NAD^+^ precursor. NRH boosts NAD^+^ levels via a proposed alternative salvage pathway of reduced intermediates using adenosine kinase (ADK) [13,14]. NADP^+^ can be formed by NAD^+^ kinases (NADK) from NAD^+^. In addition, NADP^+^ can be produced from NADPH-dependent enzymes such as glutathione reductase [15]. Several catabolites of both Nam and NA have been described such as methylated and oxidation products. *N*-methyl-nicotinamide (MeNam), the pyridones *N*-methyl-2-pyridone-5-carboxamide (Me2PY) and *N*-methyl-4-pyridone-3-carboxamide (Me4PY), and nicotinamide-*N*-oxide (NNO) have been reported in urine of subjects upon Nam ingestion while nicotinuric acid (NUA) is a specific catabolite of NA [16].

Many methods to assess levels and ratios of redox cofactors in biological systems have been described including traditional enzymatic assays (colorimetric, fluorometric). Redox cofactors require specific sample preparation methods, but even then, certain assays are often unable to distinguish specific forms. Genetically encoded sensors of NAD(P)(H) exist and are able to provide compartment-specific resolution of these cofactors in cells [17]. However, these new methods are unable to monitor several biosynthetic intermediates in parallel. Nuclear magnetic resonance (NMR) and mass spectrometry (MS) are better suited for metabolomic studies since several intermediates can be monitored simultaneously [18]. ^31^P NMR [19] has been classically used to measure phosphate-containing redox cofactors and nucleotide phosphates, and more recently, the potential of ^1^H NMR has been applied to redox coenzymes [20,21]. Liquid chromatography (LC) MS has a relatively higher sensitivity while being able to quantitatively monitor many intermediates at the same time. Different separation methods compatible with MS have been used for monitoring polar metabolites including pyridinium nucleotide cofactors. These methods include (i) the addition of an ion pairing agent in reversed-phase LC [22,23], (ii) ion exchange LC [24], (iii) supercritical fluid chromatography [25] and (iv) hydrophilic interaction liquid chromatography (HILIC) [23,26]. HILIC offers advantages since regular LC hardware and mild conditions are used without contamination from additives used in other methods such as ion pairing agents.

The reactivity of redox cofactors represents a challenge in obtaining robust direct measurements in biological matrices, so sample harvest and preparation are crucial for their appropriate measurement [23]. The use of quenching protocols and internal standards are essential in sample preparation. Furthermore, pre-analytical procedures [27,28], including harvest, storage and freeze–thaw cycles affect the analysis of the NAD^+^ metabolome. Hence, dedicated protocols need to be implemented for research and clinical applications.

We established a method for the direct measurement of the NAD^+^ metabolome in a variety of biological samples from cell cultures to rodent tissues and biofluids to human biofluids. Our method is based on HILIC-LC-MS/MS and it measures most metabolic intermediates of NAD^+^ biosynthesis, including the redox cofactors NAD(P)(H), their precursors and catabolites, a total of 18 metabolites. A specific quantitative clinical application method was also established. Additionally, the major limitations in the analysis of the NAD^+^ metabolome in whole blood of free-living population are discussed.

## 2. Results and Discussion

### 2.1. Setting up the NAD^+^ Metabolome LC-MS/MS Method

The aim of this study was to develop a method to directly monitor metabolic intermediates of the NAD(H) biosynthesis (Figure 1, Table 1) in biological samples. Given the polarity of the included metabolites, HILIC was selected as a separation method coupled to MS detection. HILIC separates very polar analytes such as the ones in Table 1 without addition of chemical modifiers. For example, the predicted distribution coefficient, log D, of NADP^+^ at pH 9 is −16.24 [30]. In addition, all metabolites vary in hydrophilicity despite being pyridine derivatives with molecular masses ranging between 122 (Nam) and 745 (NADPH) Da. Some metabolites are zwitterionic, like NA, which offered chromatographic challenges, necessitating the addition of standards to the biological matrix for confirming retention time.

MS detection was optimized in positive ion mode for each metabolite in a triple quadrupole MS, leading to two pairs of molecular ion/product ion (obtained by multiple reaction monitoring, MRM) per metabolite (Table 1). The product ions obtained were in accordance with those reported in literature [31]. For instance, the fragmentation pattern of NADP^+^ (744 *m*/*z*) led to the product ion 604 *m*/*z* corresponding to the loss of nicotinamide and water, and 508 *m*/*z* corresponding to ADP phosphate. Me2PY and Me4PY, which are regioisomers, could not be separated in this set-up since they share the same mass and retention time. Hence, MeXPY is used to for one or both metabolites.

Since HILIC-MS may be more prone to matrix effects compared to other chromatographies [32], seven different isotopically labeled internal standards (deuterated or ^13^C-labeled—Table 1) were used to account for variation (including sample preparation) during the analysis. Based on distinct retention times, mass and fragmentation patterns, a total of 18 metabolites—the NAD^+^ metabolome—were separated and detected by HILIC-based LC-MS/MS within a total run time of <20 min. A representative chromatogram of the detected metabolites is included in the Appendix A.

### 2.2. NAD^+^ Metabolome Sample Preparation

The (bio)chemical instability of redox cofactors is well known and has been recently documented in metabolomics approaches [23]. Thus, arresting metabolism during sample harvest and maintaining it during sample preparation is required for accurate readouts of these metabolites. This is accomplished with a quenching step followed by liquid extraction with organic solvent(s). For all biological samples tested in this study, the quenching step involved arresting metabolism with liquid nitrogen before storage at −80 °C until analysis. This is a well-accepted strategy when dealing with biological samples [33]. To assess the performance of sample extraction before NAD^+^ metabolome analysis, three different procedures were tested on intracellular contents of Hep G2 cells. The liquid or liquid–liquid extractions procedures were chosen based on their reported efficiency in metabolomics analysis and specifically NAD(P)(H) species [23,34]: (i) 80% cold methanol, (ii) 40:40:20 with 0.1 M formic acid and (iii) biphasic extraction. All three procedures apply cold conditions, either dry ice or ice, during organic solvent addition to preserve metabolite integrity.

The sample preparation procedures were compared for coverage and reproducibility on a subset of relevant NAD^+^ metabolites listed in Figure 2 and Appendix A. All procedures were equivalent in terms of coverage: Nam, NR, NMN, NADP^+^, NADPH, NAD^+^ and NADH were detected in all extractions. However, the biphasic extraction had slightly higher levels for most metabolites tested, specifically the redox cofactors, compared to the other two procedures. The three procedures showed comparable performance with an average of 12% error. The biphasic extraction was then adopted for all other biological samples tested in this study, so that procedures could be conducted in a single workflow.

To assess the stability of the NAD^+^ metabolites in solution and in particular the effect of delay times in the autosampler before analysis, we tested the same calibration solutions after 24 h at 4 °C (autosampler temperature). Most metabolites proved to be stable (<10%) during this time frame: NADP^+^, NADPH, NMN, NAD^+^ and NADH (Appendix A). As a general approach, however, standard solutions should be prepared as fresh as possible, avoiding freeze–thaw cycles, and stored at −80 °C to avoid possible oxidation and/or degradation over time.

### 2.3. NAD^+^ Metabolome Is Sensitive to Precursor Administration

To test the applicability of the method to monitor perturbations in NAD^+^ biosynthesis, the precursors Nam, NR and NRH were added to Hep G2 cells in culture and their intracellular contents subsequently measured using our optimized method (Figure 3 and Appendix A). All precursors were internalized by the cell and led to increased levels compared to control. Different metabolites in the NAD^+^ subnetwork were modulated by the addition of these precursors, indicating the applicability of our method in detecting perturbations in these metabolic pathways. Under these conditions, all the precursors led to increased levels of NAD^+^. NRH increased NAD^+^ levels the most (15-fold) compared to other precursors and also increased other intermediates in the NAD^+^ biosynthesis in accordance with that previously reported in an in vivo model [13]. In comparison, NR led to a modest increase in NAD^+^ probably due to lower expression of NRK1 in this cell line [35,36].

### 2.4. NAD^+^ Metabolome across Pre-Clinical Samples

To assess the applicability of our extraction procedure to pre-clinical samples, murine tissues and biofluids were tested (Appendix A). Thirteen out of eighteen NAD^+^ intermediates were detected in the samples tested, indicating that the sample preparation was adequate for different matrices. Analyses of the NAD^+^ metabolome in murine samples have been previously described by us [13,37] and others [23,38]. In the absence of stimulation with NAD^+^ precursors, most metabolites were detected in tissues although fewer metabolites were detected in plasma and urine. Neither plasma nor urine showed detectable levels of redox cofactors NAD(P)(H). Interestingly, endogenous levels of NR and NRH were detected in plasma, urine and most tissues. NRH was also reported endogenously in murine liver by others [14]. In sum, a wide range of NAD^+^ metabolites can be covered in pre-clinical tissues, even in the absence of NAD^+^ precursor stimulation. Additional NAD^+^ metabolites may be detected in other samples and conditions particularly after NAD^+^ precursor addition to animals.

### 2.5. NAD^+^ Metabolome in Clinical Samples

The endogenous presence of NAD^+^ metabolites were tested in human urine and blood samples (Table 2). Few metabolites were detected in urine, plasma and serum, but 12 out of the 18 metabolites were detected in whole blood. The precursor NRH was found in all the human biofluids tested. For example, ~1.5 µM of NRH was detected in human urine (a pool from at least two donors, two replicates) (Appendix A).

Blood samples are generally used for clinical applications. Thus, to set up a quantitative NAD^+^ metabolome method for clinical applications, several pre-analytical aspects were tested: (i) different human blood matrices were compared for metabolite coverage, (ii) the effect of aliquot volume and (iii) the number of freeze–thaw cycles on metabolite stability. Lastly, a quantitative NAD^+^ metabolome method on human whole blood was implemented.

#### 2.5.1. Selection of Blood Matrix

To apply our workflow to clinical samples, human blood matrices were compared for the presence of NAD^+^ metabolic intermediates. Human plasma, serum and whole blood from the same blood draw of nine volunteers were analyzed for their NAD^+^ metabolome (Table 2). NA, NUA, NNO, NAR, NAAD, NAMN and NMNH were not detected in any of the blood matrices in samples from a free-living population without prior exposure to exogenous NAD^+^ precursors. As expected, NAD^+^, NADH, NADP^+^ and NADPH were undetectable in plasma. These compounds were only found in whole blood, confirming that the presence of these metabolites results from blood cellular contents. These findings are in agreement with previous reports in which NAD^+^ was detected in whole blood [21,28,39,40], red blood cells (RBCs) [41] and peripheral blood mononuclear cells (PBMCs) [38], but not in plasma. Whole blood is ~45% RBCs and ~1% white blood cells with the remaining volume as plasma.

These results suggest that our workflow is amenable to detect the NAD^+^ metabolome in the blood of free-living populations and that whole blood is the best matrix for clinical samples. Using whole blood avoids isolation of cell populations, such as RBCs or PBMCs, from larger volumes [38]. This circumvents additional steps, preventing possible metabolite degradation. Hence, whole blood is a useful matrix to assess cellular states in health and disease.

#### 2.5.2. Effect of Aliquot Volume at Blood Draw

The effect of aliquot volume was assessed since larger sample volumes take longer to thaw before subsequent sample preparation. Aliquots of whole blood from the same blood draw were obtained from three volunteers with volumes between 0.1 and 4 mL. All samples were snap-frozen in liquid nitrogen. Most metabolites (MeXPY, trigonelline, NADP^+^ and NMN) showed stable signals (Appendix A), across aliquot volumes. NAD(H) and NADPH showed a tendency to decrease with increasing aliquot volume (<20%), while Nam increased (25%) (Figure 4A). These results suggested that NAD^+^ degrades to Nam in whole blood, a phenomenon previously documented [28]. Given this outcome, aliquot volume of whole blood is a determinant in the assessment of the NAD^+^ metabolome: samples volumes should be as small as required for all planned analytical analyses.

#### 2.5.3. Effect of Freeze–Thaw Cycles

Triplicate analyses of the same aliquot of human whole blood (1 mL) were performed after three freeze–thaw cycles and NAD^+^ metabolome subsequently measured. While certain metabolites proved to be stable over three freeze–thaw cycles (Appendix A), NAD^+^ and Nam suffered dramatic changes (Figure 4B). NAD(H) decreased while Nam values increased. Notably, <20% of NAD(H) were recovered after the second freeze–thaw cycle and >93% of NAD(H) disappeared after the third cycle. This is matched with a 3- and 4-fold increase of Nam on the second and third freeze–thaw cycles, respectively. It is therefore crucial to avoid freeze–thaw cycles during analysis of NAD^+^ metabolome in whole blood. These results are in agreement with the findings from analysis of aliquot volumes (above).

#### 2.5.4. Setting up a Whole Blood Quantitative NAD^+^ Metabolome Method

Trig, NRH, NR and MeNam were either just above detection limits or not consistently detected in the whole blood of free-living population using the current LC-MS/MS method; these metabolites were therefore excluded from the quantification procedure. Linearity between metabolite concentrations and instrumental response (*r*^2^ > 0.98) was obtained between 0.3 and 20 μM for all metabolites using a linear regression (Table 2, Appendix A). The limits of detection (LOD) and quantification (LOQ) characterized as signal to noise ratios (S/N) >3 and >10, respectively, were assessed (Table 2). In essence, the method presented here led to comparable LOQs of previously reported methods developed for cell systems [26,42] (sub- to micromolar ranges). Our method took advantage of using multiple internal standards, achieving dedicated LC-MS/MS parameters for each metabolite, to be applied for the analysis of human whole blood samples.

Using this quantitative method, levels of NAD^+^ metabolome in nine human volunteers were quantified (Table 2, Appendix A). MeXPY showed concentrations above the upper limit of quantification (>20 μM) for all subjects. The inter-individual variation in NAD^+^ metabolome was considerable for this group of uncharacterized subjects (e.g., the range for NADPH was 80%). For NAD^+^, the concentrations obtained were on the lower side (10–18 μM) of the range obtained by others in whole blood (16–42 μM) [21,28,39,40]. 

The reported variability (biological and/or instrumental) for redox metabolite ratios in whole blood is extensive. For example, NAD^+^/NADH ratios of 1.3–34 and 0.4–4.0 for NADPH/NADP^+^ have been described [21,40]. Our results fit in the lower end of these intervals. Redox ratios in whole blood containing RBCs are likely to be different compared to other cell types which are highly influenced by the status of surrounding tissues and their cellular states. NAD^+^ is the more biochemically and thermodynamically favorable species (compared to NADH), and also the most abundant redox metabolite in the cell. NAD^+^ and NADH can be found at the micromolar range. NAD^+^/NADH ratios are known to vary among conditions (diet, growth conditions, diseases), in different cells or tissue types with estimated reported values between 1.3 and 700 [21,23,43]. Likewise, the NADPH/NADP^+^ ratios may also vary extensively since NADPH is normally at a higher concentration than NADP^+^ in the cell. Ratios of NADPH/NADP^+^ have been reported between 0.4 and 200 among different cell and tissue types [21,23,43].

The fact that most studies reporting NAD^+^ levels in humans include a small number of subjects (<15 individuals) [21,28,40,41] makes it difficult to compare the NAD^+^ metabolome levels across studies and the ability to assess population ranges for these metabolites. In addition, different (or even uncharacterized) conditions were used regarding, for example, blood collection, analytical procedures, population characteristics—age, body mass index, nutritional or metabolic status/condition, and other factors.

In sum, our quantitative NAD^+^ method on human whole blood allowed for the coverage and quantitation of the NAD^+^ metabolome beyond NAD^+^. Pre-analytical conditions, such as avoiding freeze–thaw cycles, remain important to use a workflow that can be applied to clinical samples from observational and/or intervention studies. In the analysis of large clinical studies, other parameters may be further assessed to ensure appropriate validation of the analytical procedure for this purpose, such as inter-day variation and matrix effects.

## 3. Materials and Methods

### 3.1. Chemicals

Ammonium hydroxide solution (28% NH_3_ in H_2_O, ≥99.99% purity), ammonium bicarbonate (99–101% purity) and sodium chloride were purchased from Sigma-Aldrich (St Louis, MO, USA). HiPerSolv CHROMANORM for LC-MS acetonitrile (≥99.9% purity) was obtained from VWR Chemicals (Lutterworth, UK), and LiChrosolv hypergrade for LC-MS methanol and chloroform were obtained from Merck (Darmstadt, Germany). ULC/MS grade formic acid (≥99% purity) and ammonium acetate (≥99% purity) were obtained from Biosolve (Valkenswaard, The Netherlands). Ultrapure water was obtained from a water purification unit (Millipore). High glucose Dulbecco’s Modified Eagle Medium (D-MEM), penicillin–streptomycin, dialyzed fetal bovine serum (FBS), l-glutamine, and non-essential amino acids were purchased from Thermo Fisher, Life Technologies (Bleiswijk, the Netherlands).

### 3.2. Standard Compounds

Nicotinamide (≥99.5% purity), nicotinic acid (99.5% purity), trigonelline hydrochloride (analytical standard), β-nicotinamide adenine dinucleotide reduced disodium salt hydrate (≥97% purity), nicotinic acid adenine dinucleotide sodium salt (≥98% purity), β-nicotinamide adenine dinucleotide phosphate reduced tetra(cyclohexylammonium) salt (≥93% purity), nicotinic acid mononucleotide (≥98% purity), β-nicotinamide adenine dinucleotide hydrate (≥99% purity), β-nicotinamide adenine dinucleotide phosphate hydrate (98% purity), β-nicotinamide mononucleotide (98.5% purity), 1-methylnicotinamide chloride (≥98% purity), nicotinamide impurity E (nicotinamide *N*-oxide, 98.5% purity) and nicotinuric acid (≥98% purity) were purchased from Sigma-Aldrich/Merck (St Louis, MO, USA). Nudifloramide, 1,4-dihydro-1-methyl-4-oxo-3-pyridinecarboxamide, and nicotinic acid riboside were obtained from Toronto Research Chemicals (Toronto, Ontario, Canada). Nicotinamide riboside chloride was obtained from Chromadex (Irvine, CA, USA). The following isotopically labeled materials were used as analytical internal standards: Nicotinamide-2,4,5,6-d_4_ (98 atom % D, Nam-d_4_) and trigonelline-d_3_ HCl (*N*-methyl-d_3_, Trig-d_3_) purchased from CDN Isotopes (Québec, Canada), nicotinamide riboside-d_4_ triflate (d_3_-Major) α/β mixture (NR-d_4_) purchased from Toronto Research Chemicals (Toronto, Canada) and in-house produced U-^13^C-biomass (yeast extract), allowing for detection of ^13^C-NAD(P)(H).

Reduced nicotinamide riboside (or 1-(beta-D-ribofuranosyl)-1,4-dihydronicotinamide) was obtained by organic synthesis, as reported previously [13,44]. A sodium salt of reduced nicotinamide mononucleotide (NMNH, sodium ((2*R*,3*S*,4*R*,5*R*)-5-(3-carbamoylpyridin-1(4*H*)-yl)-3,4-dihydroxytetrahydrofuran-2-yl)methyl phosphate) was obtained by chemical synthesis using the following procedure: nicotinamide mononucleotide (0.163 g; 0.5 mmol) was dissolved in water (2 mL) under nitrogen atmosphere and this solution was cooled in an ice bath. Saturated aqueous NaHCO_3_ solution (4 mL) was added while stirring, followed by addition of sodium dithionite (85% Na_2_S_2_O_4_; 0.30 g; 1.5 mmol; 3 equiv.) in one single portion. The reaction solution was stirred while cooling for ca. 2–2.5 h. An additional portion of Na_2_S_2_O_4_ (0.13 g) was then added to the reaction solution and the reaction was continued for another hour. The solution was transferred into a freezer (−80 °C) and kept overnight. A pale-yellowish solid product (0.67 g; quantitative yield considering inorganic salts present in the product) was obtained after freeze-drying. The product was characterized by NMR (see Appendix A) and MS: ^31^P NMR (161.97 MHz, D_2_O), δ, ppm: 3.77; ^1^H NMR (400.11 MHz, D_2_O), δ, ppm: 2.98–2.99 (m, 2H, H4), 3.77–3.85 (m, 2H, H5′), 4.00–4.01 (m, 1H, H4′), 4.15 (dd, 1H, *J*_HH_= 2.3 Hz, *J*_HH_= 5.4 Hz, H3′), 4.23 (dd, 1H, *J*_HH_ = 7.2 Hz, *J*_HH_ = 5.6 Hz, H2′), 4.81 (d, 1H, *J*_HH_ = 7.3 Hz, H1′), 4.95 (dt, 1H, *J*_HH_ = 3.4 Hz, *J*_HH_ = 8.2 Hz, H5), 6.14 (dd, 1H, *J*_HH_ = 1.4 Hz, *J*_HH_ = 8.2 Hz, H6), 7.07 (s, 1H, H2). ^13^C NMR (100.61 Hz, D_2_O), δ, ppm: 21.99 (C4), 64.03 (d, *J*_CP_ = 4.2 Hz, C5′), 70.65 (C2′, C3′), 83.04 (d, *J*_CP_ = 8.6 Hz, C4′), 94.97 (C1′), 100.75 (C3), 105.45 (C5), 124.96 (C6), 138.26 (C2), 177.03 (CO). MS: found *m*/*z* = 359.38 [M+Na]^+^, 381.01 [M+2Na–H]^+^.

### 3.3. Mammalian Cell Culture

The human liver cell line Hep G2 (purchased from Sigma-Aldrich) was used as a cell model to test extraction performance and perturbation effects of NAD^+^ precursors on the NAD^+^ metabolome. Cells were grown in a 6-well format (9 cm^2^ of surface area/well) under a humidified atmosphere with 5% CO_2_ at 37 °C in high glucose DMEM medium without glutamine and without pyruvate (Thermo Fisher), supplemented with 1% penicillin-streptomycin (Thermo Fisher), 10% FBS (Thermo Fisher), 1% glutamine (200 mM, Thermo Fisher), 1% sodium pyruvate (100 mM, Thermo Fisher) and 1% MEM non-essential amino acids (Thermo Fisher). For the analysis of endogenous NAD^+^ metabolism intermediates, 500,000 cells were seeded per well in 6-well plates and harvested after overnight culture. For the analysis of NAD^+^ metabolism intermediates upon treatment with different precursors, 300,000 cells were seeded per well in 6-well plates and after 1-day culture, a medium change with the different precursors was performed. After another 20 h, cells were harvested. For harvest, cells were washed with 0.9% (*m*/*v*) sodium chloride and after discarding the washing solution, the 6-well plate was transferred into liquid nitrogen. After this quenching step, plates were stored at −80 °C until extraction.

### 3.4. Animal Tissues

C57BL/6 mice were kept in a temperature and humidity-controlled environment with a 12:12 h light–dark cycle, having access to nesting materials and were provided with ad libitum access to water and food. Further details on animal care were described elsewhere [37]. All tissues were flash-frozen in liquid nitrogen, immediately after harvest, and stored at −80 °C prior analysis.

### 3.5. Clinical samples

#### 3.5.1. Human Blood (Whole Blood, Plasma, Serum)

Donors were anonymously recruited in fasting conditions by the Metabolic Unit at Nestlé Research (Vers-chez-les-Blanc, Switzerland). For the first pilot study, plasma, serum and whole blood were collected from the same blood draw from 9 volunteers. Whole blood (~5 mL) was collected in vacutainer tubes and aliquots of 60 µL of whole blood (flash-frozen in liquid nitrogen within 10 min of collection), plasma (collecting tubes with EDTA were used, and centrifuged twice at 4 °C) and serum (collecting tubes without anticoagulant were used, and aliquots were left for 40 min at room temperature before centrifugation). All samples were flash-frozen in liquid nitrogen and stored at −80 °C prior analysis. For the second pilot study, whole blood was collected from 3 donors to assess effect of aliquot volume and freeze–thaw cycles. From each blood draw, five aliquots of different volumes were rapidly transferred in cold conditions, (4 °C) to individual tubes (4, 2, 1, 0.5 and 0.1 mL) in a maximum of 10 min time and subsequently flash-frozen in liquid nitrogen and stored at -80°C. To assess the effect of freeze–thaw cycles, the same 1 mL aliquot was thawed three times for analysis (days 1, 13 and 21).

#### 3.5.2. Human Urine

Urine (Seronorm) from at least 2 donors was purchased from Sero (Billingstad, Norway) and kept at −80 °C before analysis.

### 3.6. Sample Extraction

Different sample preparation methods were used for the analysis of the NAD^+^ metabolome. The first two methods (80% cold methanol and 40:40:20 with 0.1 M formic acid) were based on Lu et al. [23] and the third one was based on Elia et al. [34]. All samples were randomized before analysis.

#### 3.6.1. Eighty Percent Cold Methanol (‘80MeOH’)

Previously quenched and frozen Hep G2 cells were extracted with 800 µL of pre-cooled on dry ice solution of 80% (*v*/*v*) methanol in water, while adding 5 µM Nam-d_4_ and 60 µL of U-^13^C-biomass, as internal standards. The cells were scraped, transferred into tubes, vortexed for 10 s and incubated on dry ice for 20 min. The extracts were centrifuged for 15 min at 21,130× *g* at 4 °C, and the supernatant recovered and evaporated in a vacuum centrifuge at 4 °C and 5 mbar. The dried extracts were stored at −80 °C, before analysis. Just before LC-MS/MS analysis, the samples were reconstituted in 25 µL of 60% (*v*/*v*) acetonitrile/water and transferred into glass vials.

#### 3.6.2. Volume Proportion 40:40:20 with 0.1M Formic Acid (’40:20:20′)

Previously quenched and frozen Hep G2 cells were extracted with 800 µL of pre-cooled on ice solution of 40:40:20 (*v*/*v*) acetonitrile–methanol–water with 0.1 M formic acid, while adding 5 µM Nam-d_4_ and 60 µL of [U-^13^C]-biomass, as internal standards. The cells were scraped, transferred into tubes, vortexed for 10 s and incubated on ice for 3 min. An aliquot (69.6 µL) of ice-cold 15% (*m*/*v*) ammonium bicarbonate was added to the extracts for neutralization, followed by 20 min incubation on dry ice. The extracts were centrifuged, the supernatants recovered and dried, and reconstituted for LC-MS/MS analysis as described in the 80% cold methanol procedure.

#### 3.6.3. Biphasic Extraction (‘BE’)

Frozen biological materials (tissues, biofluids and cells) were extracted using an adapted protocol from Bligh and Dyer, with 1300 µL cold methanol–water–chloroform 5:3:5 (*v*/*v*) while adding 10 µM Nam-d_4_, NR-d_4_, and Trig-d_3_ and 60 µL of [U-^13^C]-biomass, keeping the samples cold throughout the procedure. For biofluids, 60 µL were taken for extraction (plasma, serum, whole blood and urine). For tissues, 2–10 mg of tissue were homogenized using 3 mm tungsten carbide beads using a tissue grinder (Qiagen TissueLyser II, Hombrechtikon, Germany) for 3 min at 20 Hz. Mammalian cells were scraped with extraction solution with a 1 mL-pipet tip, well by well, and transferred into tubes. All samples followed 10 min of agitation at 4 °C in a thermo-shaker (Thermomixer C, Eppendorf), followed by 10 min centrifugation at 21,130× *g*. After extraction, the upper polar phase was recovered, as well as the protein layer taken for total protein quantification (cells, kidney and liver) by bicinchoninic acid (BCA) assay (ThermoFisher Scientific). The recovered extract was dried overnight in a vacuum centrifuge at 4 °C and 5 mbar, and then stored at −80 °C, before analysis. Dried samples were reconstituted in either 25 µL (cells) or 60 µL (whole blood, plasma, serum, urine, liver, kidney, pancreas and muscle) in 60% (*v*/*v*) acetonitrile/water, and the supernatants transferred into glass vials for LC-MS/MS analysis.

### 3.7. Liquid Chromatography Mass Spectrometry

The UHPLC consisted of a binary pump, a cooled autosampler, and a column oven (DIONEX Ultimate 3000 UHPLC+ Focused, Thermo Scientific), connected to a triple quadrupole mass spectrometer (TSQ Vantage, Thermo Scientific) equipped with a heated electrospray ionization (H-ESI). Chromatographic conditions were based on reported metabolomics analysis of polar metabolites [45]. In detail, 2 µL of each sample were injected into the analytical column (2.1 mm × 150 mm, 5 µm pore size, 200 Å HILICON iHILIC^®^-Fusion(P)), guarded by a pre-column (2.1 mm × 20 mm, 200 Å HILICON iHILIC^®^-Fusion(P) Guard Kit) operating at 35 °C. The mobile phase (10 mM ammonium acetate at pH ~9, A, and acetonitrile, B) was pumped at a flow rate of 0.25 mL/min flow over a linear gradient of decreasing organic solvent (0.5–17 min, 92.5–25% B), followed by washing (17–21 min at 25% B) and re-equilibration (21.1–30 min at 92.5% B). Retention time of individual analytes was confirmed using authentic standards. MS conditions were obtained by single compound optimization, via infusion of standard compounds into the MS source, to obtain specific parent and product ion pairs (multiple reaction monitoring, MRM) per metabolite. In the final method, the MS operated in positive mode ionization at 3500 V on MRM. The scan width was 1 *m*/*z* for each scan and scan time was 0.05 s, the peak width for Q1 was 0.25 FWHM and for Q3 0.70 FWHM. The sheath gas was set to 20 arbitrary units, and the auxiliary gas was kept at 15 arbitrary units. The vaporizer temperature was set to 280 °C and the temperature of the ion transfer tube to 310 °C. The tube lens (S-lens) and collision energy were individually optimized for each analyte, using a 0.05 s scan time.

### 3.8. Data Analysis

The software Xcalibur v. 4.1.31.9 (Thermo Scientific) was used for instrument control, data acquisition and data processing. Positive ion mode extracted chromatograms using the metabolites MRM traces were integrated, normalized to internal standard (plasma, blood, urine, kidney, liver, cells) and total protein (kidney, liver, cells). Retention time and mass detection of metabolites were confirmed with authentic standards. For the quantification of the NAD^+^ metabolome in whole blood, calibration curves were constructed from the analysis of 10 standard compound aqueous solutions (range 0.3 to 20 μM), including internal standards, submitted to the same extraction procedure as regular samples. A linear regression was calculated per metabolite. As quantification criteria, a minimum of five points were evenly distributed over the regression line and the coefficient of determination was >0.9. Further statistics were done in R, v. 3.5.0 (R Core Team).

## 4. Conclusions

Given the pivotal role of NAD(P)(H) to maintain energy and redox balances, great interest has emerged to understand NAD^+^ biology in health and disease, from metabolic to signaling phenomena. We developed and presented a method with widespread applicability. The method includes 18 NAD^+^ metabolites and it was tested on a variety of biological samples, from cell lines to mouse tissues and biofluids, and human samples.

Sample acquisition and handling can alter measurement of redox cofactors since some metabolites in the NAD^+^ metabolome measurements are labile in human whole blood. Thus, pre-analytical and analytical protocols must be rigorously established for the specific biological matrices of interest to prevent potential degradation and ensure robust analytical outcomes.

The coverage of the NAD^+^ metabolome and the concentrations of the metabolites are dependent on sample type (tissue, biofluid) and treatment/condition/nutritional status. Nevertheless, to better understand the dynamic behavior of the NAD^+^ subnetwork, a wide metabolite coverage is essential to map pathways and the metabolic fate of specific precursors.

Finally, being able to monitor the NAD^+^ metabolome at the clinical level becomes attractive not only for assessing NAD^+^-related metabolic status in populations, but also for interindividual variability and to evaluate therapies with NAD^+^ precursors.

## Figures and Tables

**Figure 1 ijms-22-10598-f001:**
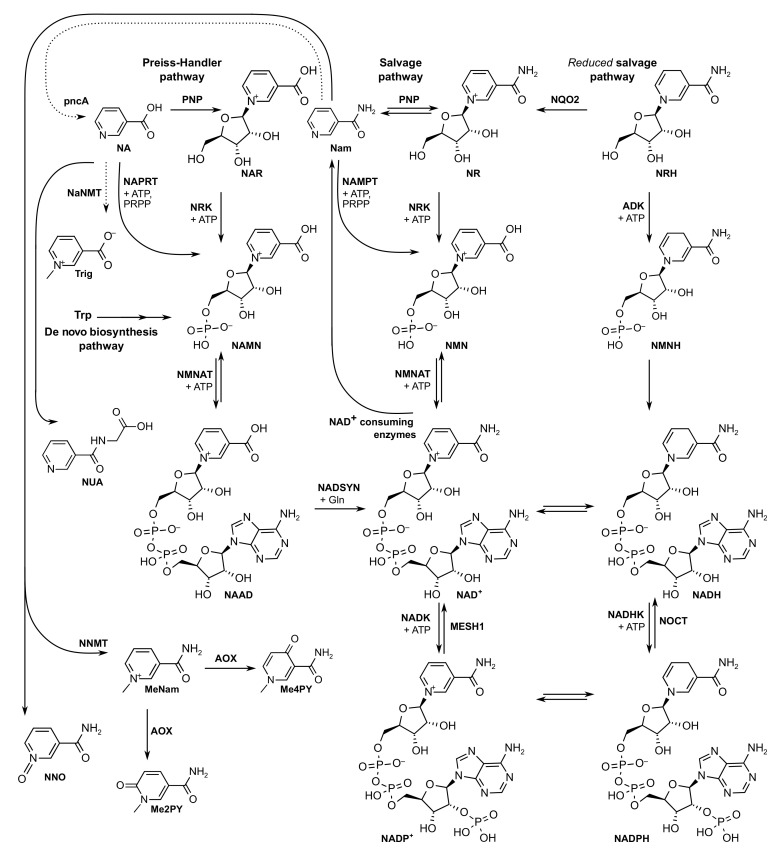
The NAD^+^ metabolome of biosynthetic pathways of NAD(P)(H), including all metabolites monitored in this study. Metabolites with chemical structures are monitored while those labeled only with acronyms were not monitored in this method. NAD^+^ consuming enzymes include specifically: sirtuins, ADP-ribosyl cyclases, SARM or PARPs. All enzymatic steps were found in mammalians unless indicated by dashed arrows, according to KEGG [29]. Metabolites with structure: NA: nicotinic acid; NAR: nicotinic acid riboside; Nam: nicotinamide; NR: nicotinamide riboside; NRH: nicotinamide riboside, reduced form; NAMN: nicotinic acid mononucleotide; NMN: nicotinamide mononucleotide; NMNH: nicotinamide mononucleotide, reduced form; NAAD: nicotinic acid adenine dinucleotide; NAD^+^: nicotinamide adenine dinucleotide; NADH: nicotinamide adenine dinucleotide, reduced form; NADP^+^: nicotinamide adenine dinucleotide monophosphate; NADPH: nicotinamide adenine dinucleotide monophosphate, reduced form; Trig: trigonelline; MeNam: *N*-methylnicotinamide; Me2PY: *N*-methyl-2-pyridone-5-carboxamide; Me4PY: *N*-methyl-4-pyridone-3-carboxamide; NUA: nicotinuric acid; NNO: nicotinamide-*N*-oxide. Metabolites without defined chemical structures (not monitored): Trp: tryptophan; Gln: glutamine; ATP: adenosine triphosphate; PRPP: 5-Phosphoribosyl 1-pyrophosphate. Enzymes: pncA: Nicotinamidase; PNP: purine nucleosidase; NQO2: *N*-ribosyldihydronicotinamide quinone reductase 2; ADK: adenosine kinase; NAPRT: NA phosphoribosyltransferase; NAMPT: Nam phosphoribosyltransferase; NRK: NR kinase; NMNAT: NMN adenyltransferase; NADSYN: NAD^+^ synthetase; NADK: NAD^+^ kinase; NADHK: NADH kinase; AOX: Aldehyde oxidase; NNMT: Nicotinate *N*-methyltransferase; NaNMT: NA phosphoribosyltransferase; MESH1: NADPH phosphatase; NOCT: nocturnin.

**Figure 2 ijms-22-10598-f002:**
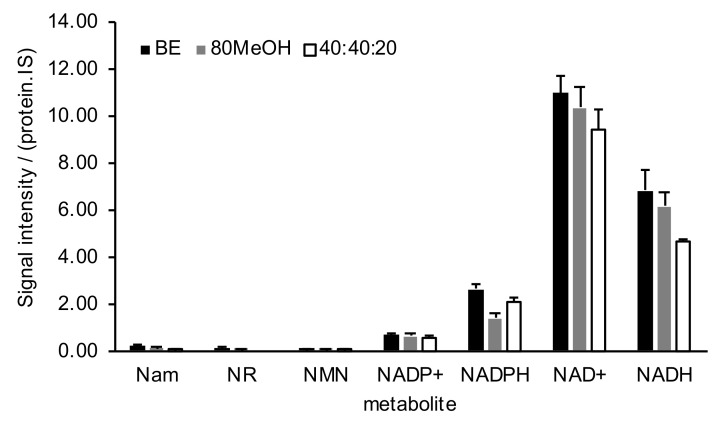
Comparison of 3 sample preparation procedures in detecting the NAD^+^ metabolome of Hep G2 cells: biphasic extraction (BE), 80% cold methanol (80MeOH) and 40:40:20 with 0.1 M formic acid (40:40:20) extraction (*n* = 5, average ±SD). Values expressed in ratios of LC-MS/MS signal intensity normalized to internal standard and total protein content (arbitrary units). Nam: nicotinamide; NR: nicotinamide riboside; NMN: nicotinamide mononucleotide; NADP^+^: nicotinamide adenine dinucleotide monophosphate; NADPH: nicotinamide adenine dinucleotide monophosphate, reduced form; NAD^+^: nicotinamide adenine dinucleotide; NADH: nicotinamide adenine dinucleotide, reduced form.

**Figure 3 ijms-22-10598-f003:**
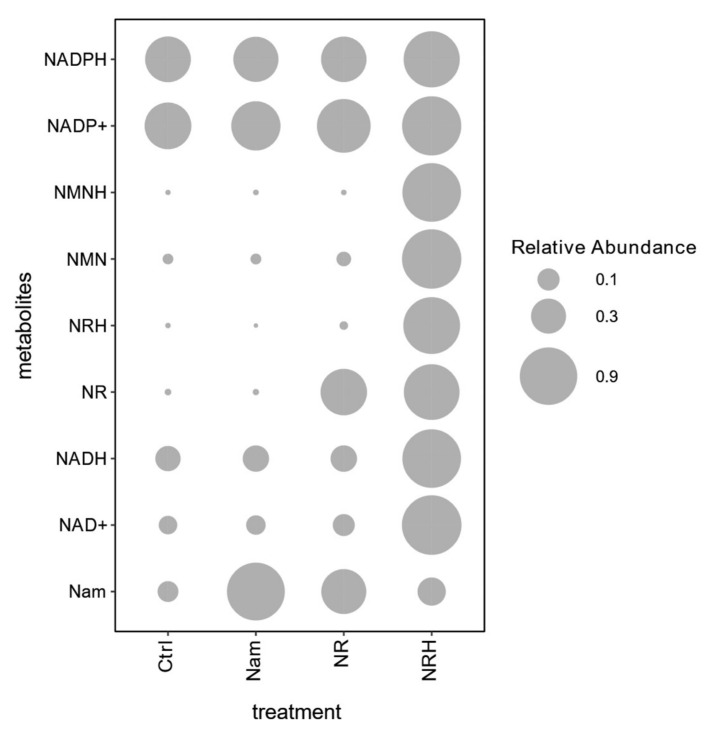
NAD^+^ metabolome of Hep G2 cells after addition of Nam, NR or NRH (0.5 mM) to the culture media and control (Ctrl). Measurements (*n* = 4) were normalized to the highest value expressed as relative abundance (log2 fold changes of average ratios of LC-MS/MS signal intensity normalized to internal standard and total protein content, in relation to control). NADPH: nicotinamide adenine dinucleotide monophosphate, reduced form; NADP^+^: nicotinamide adenine dinucleotide monophosphate; NMNH: nicotinamide mononucleotide, reduced form; NMN: nicotinamide mononucleotide; NRH: nicotinamide riboside, reduced form; NR: nicotinamide riboside; NADH: nicotinamide adenine dinucleotide, reduced form; NAD^+^: nicotinamide adenine dinucleotide; Nam: nicotinamide.

**Figure 4 ijms-22-10598-f004:**
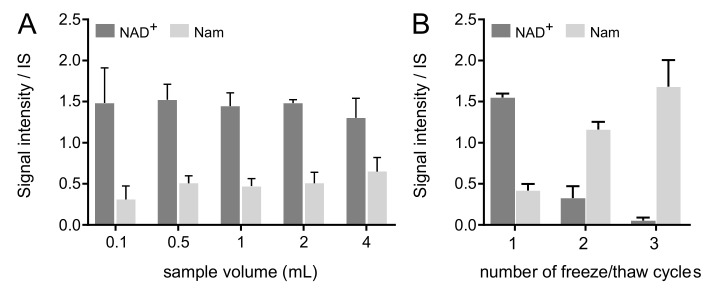
LC-MS/MS metabolite signal intensity per internal standard for NAD^+^ and Nam of human whole blood according to: (**A**), aliquot volume (0.1, 0.5, 1, 2, 4 mL) at blood draw. Average ± SD values from 3 donors (per aliquot and subject *n* = 1 for 0.1, 0.5 and 2 mL, *n* = 2 for 4 mL, and *n* = 3 for 1 mL); (**B**), number of freeze–thaw cycles (taken on 1, 13, 21 days). Average ± SD values from 3 donors (*n* = 3). NAD^+^: nicotinamide adenine dinucleotide; Nam: nicotinamide.

**Table 1 ijms-22-10598-t001:** Metabolites and LC-MS/MS settings of the NAD^+^ metabolome method.

ID	CAS nr	Rt	MF	MM	MIM	Q1 > Q3	CE	S-Lens
Nam	98-92-0	2.9	C_6_H_6_N_2_O	122.12	122.05	123 > 80, 53	26, 32	102
Nam-d_4_	347841-88-7	2.9	C_6_H_2_^2^H_4_N_2_O	126.15	126.07	127 > 84, 81	22, 18	99
MeXPY (Me2PY /Me4PY)	701-44-0, 769-49-3	3.2	C_7_H_8_N_2_O_2_	152.15	152.06	153 > 136, 92	15, 23	62
NA	59-67-6	3.9	C_6_H_5_NO_2_	123.11	123.03	124 > 78, 53	22, 31	102
NNO	1986-81-8	4.8	C_6_H_6_N_2_O_2_	138.12	138.04	139 > 106, 78	21, 31	118
NUA	583-08-4	5.3	C_8_H_8_N_2_O_3_	180.16	180.05	181 > 135, 80	16, 34	107
NRH	19132-12-8	5.4	C_11_H_16_N_2_O_5_	256.26	256.11	257 > 150, 108	24, 31	99
Trig	535-83-1	9.2	C_7_H_7_NO_2_	137.14	137.05	138 > 92, 94	21, 20	78
Trig-d_3_	-	9.2	C_7_H_4_^2^H_3_NO_2_	140.15	140.07	141 > 97, 95	21, 23	73
NAR	17720-18-2	10.5	C_11_H_13_NO_6_	255.22	255.07	256 > 124, 78	13, 40	146, 64
NADH	58-68-4	10.6	C_21_H_29_N_7_O_14_P_2_	665.44	665.12	666 > 649, 302	20, 40	119
^13^C-NADH	1259998-13-4	10.6	^13^C_21_H_29_N_7_O_14_P_2_	686.29	686.20	687 > 670, 311	20, 40	119
NMNH	4229-56-5	10.8	C_11_H_17_N_2_O_8_P	336.24	336.07	337 > 320, 108	13, 28	76
NAAD	6450-77-7	11.1	C_21_H_26_N_6_O_15_P_2_	664.41	664.09	665 > 136, 119	40	144
NAMN	321-02-8	11.4	C_11_H_14_NO_9_P	335.20	335.04	336 > 124, 97	13, 22	92
NAD^+^	53-84-9	11.5	C_21_H_27_N_7_O_14_P_2_	663.43	663.11	664 > 524, 428	20	137
^13^C-NAD^+^	1259998-09-8	11.5	^13^C_21_H_27_N_7_O_14_P_2_	684.27	684.18	685 > 539, 438	20	137
NADPH	53-57-6	11.7	C_21_H_30_N_7_O_17_P_3_	745.42	745.09	746 > 729, 428	20, 40	206
^13^C-NADPH	1259998-18-9	11.7	^13^C_21_H_30_N_7_O_17_P_3_	766.27	766.16	767 > 438, 750	20	150
NADP^+^	53-59-8	11.9	C_21_H_28_N_7_O_17_P_3_	743.41	743.08	744 > 604, 508	20	150
^13^C-NADP^+^	1259998-15-6	11.9	^13^C_21_H_28_N_7_O_17_P_3_	764.25	764.15	765 > 518, 619	20	150
NMN	1094-61-7	12.0	C_11_H_15_N_2_O_8_P	334.22	334.06	335 > 123, 80	14, 46	85
MeNam	3106-60-3	16.0	C_7_H_9_N_2_O	137.16	137.07	137 > 79, 52	31, 37	89
NR	1341-23-7	17.1	C_11_H_15_N_2_O_5_	255.25	255.10	255 > 123, 80	13, 38	64
NR-d_4_	-	17.1	C_11_H_11_^2^H_4_N_2_O_5_	259.27	259.12	259 > 127, 84	16, 34	46, 49

ID: metabolite acronym; CAS nr: Chemical Abstracts Service registry number; Rt = retention time (min); MF: molecular formula; MM: molecular mass (Da); MIM: monoisotopic mass (Da); Q1 > Q3 correspond to selected ions (*m*/*z*) in the 1st (molecular ion) and 3rd quadrupoles (product ions) of the multiple reaction monitoring; CE: collision energy (V); S-lens: tube lens or S-lens RF amplitude. Nam: nicotinamide; Me2PY: *N*-methyl-2-pyridone-5-carboxamide; Me4PY: *N*-methyl-4-pyridone-3-carboxamide; NA: nicotinic acid; NNO: nicotinamide-*N*-oxide. NUA: nicotinuric acid; NRH: nicotinamide riboside, reduced form; Trig: trigonelline; NAR: nicotinic acid riboside; NADH: nicotinamide adenine dinucleotide, reduced form; NMNH: nicotinamide mononucleotide, reduced form; NAAD: nicotinic acid adenine dinucleotide; NAMN: nicotinic acid mononucleotide; NAD^+^: nicotinamide adenine dinucleotide; NADPH: nicotinamide adenine dinucleotide monophosphate, reduced form; NADP^+^: nicotinamide adenine dinucleotide monophosphate; NMN: nicotinamide mononucleotide; MeNam: *N*-methylnicotinamide; NR: nicotinamide riboside.

**Table 2 ijms-22-10598-t002:** NAD^+^ metabolome of different human biofluids, and NAD^+^ metabolome quantification in human whole blood.

ID	Urine	Plasma	Serum	Whole Blood	Whole Blood			
					LOD	LOQ	*r* ^2^	C (Whole Blood) Min–Max ± SD
Nam	X	-	-	X	0.3	1.0	0.991	5.0 ± 0.3–10.0 ± 1.4
MeXPY	X	X	X	X	-	-	-	-
NRH	X	X	X	X	-	-	-	-
Trig	X	X	X	X	-	-	-	-
NADH	-	-	-	X	0.2	0.6	0.996	8.0 ± 1.0–16.0 ± 0.6
NADPH	-	-	-	X	0.3	1.0	0.991	7.0 ± 0.8–17.0 ± 1.0
NAD^+^	-	-	-	X	0.2	0.7	0.995	10.0 ± 1.4–18.0 ± 0.1
NADP^+^	-	-	-	X	0.3	1.1	0.989	13.0 ± 1.2–19.0 ± 5.9
NMN	-	-	-	X	0.5	1.5	0.978	2.0 ± 0.0–3.0 ± 0.2
MeNam	-	-	-	X	-	-	-	-
NR	X	-	-	X	-	-	-	-

ID: metabolite acronym; X = detected metabolite; - = not detected metabolite; LOD: limit of detection (μM); LOQ: limit of quantification (μM); *r*^2^: correlation factor obtained by 1/x^2^ weighted linear regression from calibration curve; C (whole blood): metabolite concentrations in whole blood with ranges obtained from 9 subjects (µM, 3 technical replicates). Nam: nicotinamide; MeXPY (Me2PY /Me4PY): Me2PY: *N*-methyl-2-pyridone-5-carboxamide; Me4PY: *N*-methyl-4-pyridone-3-carboxamide; NRH: nicotinamide riboside, reduced form; Trig: trigonelline; NADH: nicotinamide adenine dinucleotide, reduced form; NADPH: nicotinamide adenine dinucleotide monophosphate, reduced form; NAD^+^: nicotinamide adenine dinucleotide; NADP^+^: nicotinamide adenine dinucleotide monophosphate; NMN: nicotinamide mononucleotide; MeNam: *N*-methylnicotinamide; NR: nicotinamide riboside.

## Data Availability

Data were included in the manuscript and Appendix A.

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
