# Peer review of "A Method to Monitor the NAD+ Metabolome—From Mechanistic to Clinical Applications"

_ijms, 2021, doi:10.3390/ijms221910598_

Round 1
Reviewer 1 Report
This is a thorough presentation of the development of a method to detect the NAD+ metabolome. The authors developed the platform starting with standards and moving to cell cultures, murine biofluid and tissue extracts, and finally human biofluids and whole blood. The authors considered key factors in human sample preparation such as freeze thaw cycles, aliquot volume, and blood matrix used. Their approach is thorough. The procedures are well described and this manuscript should be of interest to readers.
Author Response
We thank the reviewer for their effort in reading our manuscript and for their appreciation of our study.
Reviewer 2 Report
Please find below my recommendations and comments:
- Please emphasize in the Introduction the total number of analytes determined using the developed method.
- I recommend using the abbreviation "LC-MS/MS" instead of "LC-MS" since the triple quadrupole mass spectrometer has been used.
- Lines 185/186: " Most metabolites proved to be stable during this time frame" - please clarify, when the analyte was considered to be stable during storage in the autosampler. The information about the established acceptable limit (e.g. +/- 10%) should be also added.
- Lines 290/291. Please highlight the advantages of the herein proposed method or improvements made in relation to available protocols.
- Section 4.7. Please explain how the settings of MS and chromatographic conditions have been optimized.
- Line 497. Please explain how total protein content was determined.
- The effect of freeze-thaw cycles should be evaluated for all biological matrices, not only for whole blood.
- Representative chromatograms of all analytes determined by the proposed LC-MS/MS method should be added.
- Validation parameters such as recovery, accuracy, precision should be determined for all studied biological matrices.
- Data about matrix effects should be added.
- Please discuss whether or not the proposed method is selective toward picolinic acid (molar mass: 123.111 g/mol) - kynurenine pathway metabolite.
- The extraction step should be optimized for all matrices studied.
- The stability of analytes in the presence of the sample matrix should be studied in parallel.
- The applicability of the proposed method to other biological samples (e.g. urine, cells, tissues) should be demonstrated.
Author Response
Reviewer 2
- Please emphasize in the Introduction the total number of analytes determined using the developed method.
- I recommend using the abbreviation "LC-MS/MS" instead of "LC-MS" since the triple quadrupole mass spectrometer has been used.
- Lines 185/186: " Most metabolites proved to be stable during this time frame" - please clarify, when the analyte was considered to be stable during storage in the autosampler. The information about the established acceptable limit (e.g. +/- 10%) should be also added.
- Lines 290/291. Please highlight the advantages of the herein proposed method or improvements made in relation to available protocols.
- Section 4.7. Please explain how the settings of MS and chromatographic conditions have been optimized.
- Line 497. Please explain how total protein content was determined.
- The effect of freeze-thaw cycles should be evaluated for all biological matrices, not only for whole blood.
- Representative chromatograms of all analytes determined by the proposed LC-MS/MS method should be added.
- Validation parameters such as recovery, accuracy, precision should be determined for all studied biological matrices.
- Data about matrix effects should be added.
- Please discuss whether or not the proposed method is selective toward picolinic acid (molar mass: 123.111 g/mol) - kynurenine pathway metabolite.
- The extraction step should be optimized for all matrices studied.
- The stability of analytes in the presence of the sample matrix should be studied in parallel.
- The applicability of the proposed method to other biological samples (e.g. urine, cells, tissues) should be demonstrated.
Reviewer 2
We thank the reviewer for their investment in reading our manuscript and their detailed comments, that enabled us to improve this new version of our manuscript.
We have numbered the reviewer’s comments for easier implementation and discussion. We have highlighted in yellow the changes performed in the manuscript.
- We have added the number of metabolites detected in our method in the introduction – L122, and also in the conclusion – L334.
- We have changed the acronym LC-MS to LC-MS/MS throughout the whole manuscript, as per the reviewer request.
- We have added a sentence on this point in the manuscript for clarity (L187-L188).
- We have added the highlights of the method and the differences to others in L294-297.
- The optimization of chromatographic (L487-488, L494-495) and MS (L495-L498) conditions were added to section 4.7.
- The total protein quantification was performed using the BCA assay, as mentioned in L476-478 (section 4.6.3).
- Pre-analytical procedures, inherent to the way samples are stored and collected are indeed important for the performance of analytical methods. However, for all other biological samples contemplated in this manuscript, these do not happen, in the sense that a sample is taken, prepared and analyzed, and thus consumed. To precise this further, a cell experiment is conducted for the purpose of specific analysis, and then totally consumed or if not totally consumed not re-frosted for other analyses; it is then discarded. For murine tissues, these are dissected and snap frozen, as mentioned in the Methods section (L422-423), and then used and discarded. For in vitro and pre-clinical analyses there is also a higher control in the way samples are collected, so the cold chain is easily maintained. Therefore, the only samples where we considered to be important to assess freeze/thaw cycles are clinical samples. In this case, sample availability may be scarce and used for different analyses, and control over sampling is more difficult, as sample collection is often taken care by nurses in a hospital setting. For this reason, this parameter was assessed in section 3.5.3, that we hope to be of use for other scientists working in this area.
We have added a sentence (L326-329) in the manuscript to clarify this point. - A chromatogram of all metabolites measured in this method was included in the supplementary materials (Fig 1S) and mentioned in the text (L155-L156).
- Validation is surely important in quantitative methods to be used at large scale. However detailed validation is mostly necessary for analytical workflows that will process many batches of samples over time. This study has as primary aim to demonstrate the wide applicability of our method to different biological sample types. It does not focus on a specific sample type to achieve optimization of large batches of samples. It is also beyond the scope to engage in validation for such a high number of biological samples analyzed, as each sample type requires its own validation procedure. This is likely to imply the use of automated sample preparation, reproducibility among analysts, etc. Yet, this method was implemented for low-scale analysis of samples at a research laboratory level. Taking this aspect into account, we have used a series of internal standards (mentioned in Table 1) that we use to correct for the analytical procedure from sample preparation to data analysis. In addition, for certain sample types, we also use protein quantification for normalization. This is important for cell samples for example. For human whole blood, pre-analytical procedures were assessed and recommended to adopt (section 3.5), that we believe will improve the robustness of this method for clinical applications.
- The matrix effects are surely important and part of the validation procedure, but unfortunately, we could not assess this point in the time frame given to perform modifications to this manuscript. We have however looked into another report using whole blood, that found that matrix effects could be compensated by the use of internal standards (Wang, D., Xu, P., and Mesaros, C. (2021). Analytical considerations for reducing the matrix effect for the sphingolipidome quantification in whole blood. Bioanalysis 13, 1037–1049). Our method contemplates various internal standards, including a 13C-NAD(P)(H) internal standards, that we hope contribute to the performance of the method. We have mentioned this point (matrix effects) as a parameter to further optimize for large scale analyses, in a sentence (L326-329) in the manuscript.
- We agree that picolinic acid would be interesting to assess, in addition to other metabolic intermediates in close proximity of NAD+. However, we have not included intermediates from the de novo biosynthesis in our study. So picolinic acid was not tested, and it will not make sense to include, without including all the other metabolites of this specific pathway. Unfortunately, this was not part of the planned study.
- The extraction was optimized for cell systems, as demonstrated in the manuscript for easy access of material and controlled conditions. This extraction proved to allow the detection of NAD+ metabolites, in a reproducible way, as well as clean-up the sample enough to allow for extended column lifetime. The same procedure enabled to robust detection of many NAD+ metabolites in all other biological tissues mentioned in this manuscript. For this reason, optimizing sample preparation for all different tissues, while it may lead to slight improvements of metabolite recovery, it may introduce bias in qualitative comparison of sample types throughout studies, if different sample preparations are adopted. In addition, different sample preparations for different tissues/sample types for the same analytical method is, as may imagine, much less practical, and thus error-prone.
We added a sentence in the manuscript to clarify this reasoning (L182-183). - We agree that stability is an important aspect in analytical methods. For this reason, the stability of standard solutions was demonstrated in this manuscript (Figure 2S), to make sure that metabolites do not degrade extensively through a batch of analysis. For cell systems and mouse tissues, the stability was not specifically assessed in this manuscript, as the batch of samples is invariably reduced, so analysis is performed immediately after extraction, as stated in the Methods section, and these can be conducted in a few days. Stability of metabolites for clinical applications is indeed an important factor, as these studies are likely to contain a higher number of samples. For this reason, we have dedicated a considerable effort on assessing freeze/thaw cycle and aliquot volume in human whole blood, that we included in this manuscript (section 3.5).
- In our study we have measured various biological samples: cells, mouse tissues and biofluids and human biofluids. So we believe to have provided data on the presence of NAD+ metabolites in all these sample types. In addition, we have proven the applicability of this method in a cell model, Hep G2, that we challenged with different NAD+ precursors (NR, Nam, NRH), highlighting that the metabolic intermediates measured showed differential levels to those in control conditions (without precursor administration) – section 3.3. In this study we analyzed the whole blood of 9 different volunteers under free living conditions, that we could quantify a subset of the NAD+ metabolome – section 3.5.4. We believe these 2 sets of experiments highlight the applicability of our method to biological applications in the current manuscript.
Proving applicability of this method on all different biological samples is beyond the scope of this study, as it would make this manuscript very lengthy. We have however made use of earlier versions of the method described here (that included less metabolites) in other studies that are already published, on other cell systems and mouse tissues and biofluids, which are referenced in this manuscript: in Sambeat et al 2019, murine livers were analyzed leading to differences in NAD+ metabolome in NRK1 LKO mice compared to control mice upon high fat diet. In Giroud-Gerbetant et al 2019, the NAD+ metabolome of AML12 hepatocytes were challenged with NRH, as well as enzyme inhibitors, leading to differences in metabolite levels, compared to control conditions. In this study, mouse plasma, liver, muscle, and kidney were also analyzed to assess impact of NRH administration on NAD+ metabolome.
Round 2
Reviewer 2 Report
Authors addressed in detail my comments and supplemeted the paper in additional information. I have no additional comments.